# Identification and Characterization of Three Novel Iflaviruses in the Cabbage Whitefly *Aleyrodes proletella*

**DOI:** 10.3390/insects16040335

**Published:** 2025-03-22

**Authors:** Zhuang-Xin Ye, Guo-Wei Gu, Peng-Peng Ren, Chuan-Xi Zhang, Jun-Min Li, Yan Zhang, Jian-Ping Chen

**Affiliations:** 1College of Forestry and Grassland, Nanjing Forestry University, Nanjing 210037, China; yzx244522794@163.com; 2State Key Laboratory for Quality and Safety of Agro-Products, Key Laboratory of Biotechnology in Plant Protection of MARA, Zhejiang Key Laboratory of Green Plant Protection, Institute of Plant Virology, Ningbo University, Ningbo 315211, China; 2011074040@nbu.edu.cn (P.-P.R.); chxzhang@zju.edu.cn (C.-X.Z.); lijunmin@nbu.edu.cn (J.-M.L.); 3Popularization of Agricultural Technical Service Station of Yuyao, Yuyao 315400, China; ggw_1983@126.com

**Keywords:** *Aleyrodes proletella*, novel viruses, *Iflavirus*, meta-transcriptomics

## Abstract

Cabbage whitefly (*Aleyrodes proletella*) is a major agricultural pest that damages crops, such as cabbage, broccoli, and kale, by feeding on plant sap. However, the virome of the *A. proletella* remains largely unexplored. In this study, we employed deep sequencing technology to investigate the virome of *A. proletella* and successfully identified the genomes of three novel viruses. Phylogenetic analysis confirmed that the three viruses represent three new species within the genus *Iflavirus* and were assigned to different branches on the phylogenetic tree. These findings enhance our understanding of viral diversity in agricultural pests and provide a foundation for future research and contribute valuable viral resources for the development of biopesticides based on viruses.

## 1. Introduction

The cabbage whitefly (*Aleyrodes proletella*), a member of the family Aleyrodidae in the order Hemiptera, is native to East, West, South and North Africa and has since spread to other regions, including South America, North America, New Zealand and Australia [1,2,3]. In New Zealand, it is widely distributed among native plants, such as New Zealand spurge (*Euphorbia glauca*) and New Zealand celery (*Apium prostratum*), as well as Brassicaceae crops (e.g., kale, cabbage, and Brussels sprouts), ornamental flowers (e.g., plants of the *Aquilegia* genus), and weeds (e.g., chicory) [4]. In addition, *A. proletella* also infests plants from the Asteraceae, Fabaceae, and Papaveraceae families in other countries [5]. *A. proletella* is a sap-sucking insect that penetrates the phloem of plants to feed on sap. The deleterious effects of *A. proletella* on crop growth primarily arises from its feeding activities, wax deposits, and the development of sooty mould due to honeydew excretion. Additionally, its egg-laying behaviour further contributes to plant stress and reduced crop yield [4,6].

With the continuous progress of high-throughput sequencing technologies, meta-transcriptomics has become an essential tool for virus discovery and identification [6,7,8]. This technology has shown great potential in studying the virome of agricultural insects, facilitating the rapid identification of novel RNA viruses in diverse insect species. For example, multiple novel RNA viruses have been identified in locusts [9], leafhoppers [10,11], planthoppers [12,13] and whitefly (*Bemisia tabaci*) [14]. Although *A. proletella* causes significant agricultural losses in many regions worldwide, its virome is still vastly understudied compared to that of *B. tabaci*. This knowledge gap limits a comprehensive understanding of viral diversity within the family Aleyrodidae. Artificial Intelligence (AI) methods and deep learning algorithms have had a significant impact on various research fields in life sciences, such as virus discovery and protein structure prediction. Recently, AlphaFold3 has been developed and applied to virus identification and viral protein structure prediction to better understand viral evolution [15].

*Iflaviridae* is a family within the order *Picornavirales* that possesses a positive-sense, single-stranded, non-segmented RNA genome (monocistronic genome) that ranges in length from approximately 9 to 11 kb. The genome encodes a single polyprotein, which is autocatalytically cleaved into structural proteins (SPs) and non-structural proteins (NSPs) involved in replication [16]. Typically, the SPs are encoded in the 5′ region of the genome, while the NSPs are encoded in the 3′ region. However, some studies have shown gene rearrangement of structural and non-structural proteins within the family *Iflaviridae*, suggesting that it might be more complex than previously understood [17]. Currently, the family *Iflaviridae* contains only one genus, *Iflavirus*, with iflaviruses identified exclusively in arthropods, primarily insects. Once the virus enters the host cell, the infection process occurs rapidly, with progeny viruses produced within hours.

In this study, we collected over 100 cabbage whiteflies from fresh leaf lettuce in Xinjiang Agricultural University and performed high-throughput sequencing. Three novel iflaviruses were identified in *A. proletella* using meta-transcriptomics for the first time, named Aleyrodes proletellus iflavirus 1 (APIV1), Aleyrodes proletellus iflavirus 2 (APIV2) and Aleyrodes proletellus iflavirus 3 (APIV3). Furthermore, the genomic structure, evolutionary relationships, and 3D structural comparisons were analyzed in detail. These findings provide a crucial foundation for understanding the composition of the virome in the *A. proletella* and exploring viral diversity within the family Aleyrodidae.

## 2. Materials and Methods

### 2.1. Sample Preparation and RNA Extraction

Cabbage whiteflies were collected from vegetable gardens of Xinjiang Agricultural University (43.80833 N, 87.56778 E, 882.3 m) on fresh leaf lettuce (*Lactuca sativa*) in June 2024 [18]. Then they were maintained in insect-proof cages on fresh romaine lettuce at 25 ± 1 °C, 50–70% relative humidity, and a 14 h light/10 h dark cycle in the laboratory of Ningbo University. Total RNA was extracted from approximately 100 whiteflies using TRIzol reagent (Takara, Beijing, China) according to the manufacturer’s protocol. The RNA quality was assessed using a NanoDrop 2000 spectrophotometer (Thermo Fisher Scientific, Waltham, MA, USA).

### 2.2. Transcriptome Sequencing

The RNA libraries were constructed using the NEBNext Ultra RNA Library Prep Kit (New England Biolabs, Ipswich, MA, USA) and sequenced on an Illumina NovaSeq platform with paired-end 150 bp reads. Raw reads were filtered to remove adaptor sequences and low-quality reads using Trimmomatic (version 0.39) [19]. De novo assembly was performed using Trinity (version 2.8.5) with default parameters. Contigs longer than 200 bp were selected for further analysis. The flited contigs were compared to the iBOL cytochrome oxidase subunit 1 (COI) database [20] using blastn to identify and filter COI sequences. The filtered COI sequence was then subjected to further analysis using the NCBI blastn tool to determine species identity.

### 2.3. Virus Discovery and Confirmation

The assembled contigs were subjected to DIAMOND (version 0.9.28.129) searches against a local viral sequence database to identify potential viral-like contigs, which consisted of representative viruses downloaded from NCBI Virus on 1 January 2025 (https://www.ncbi.nlm.nih.gov/labs/virus/vssi/#/) with an E-value cutoff of 1 × 10^−5^. To eliminate false positives, clean reads were mapped back to the viral genome to assess the sequencing depth of the viral sequence; the candidate viral contigs were further compared with the NCBI complete nucleotide (NT) and non-redundant (NR) protein databases [21].

### 2.4. Structural and Genomic Analysis

Open reading frames (ORFs) of the candidate viruses were comprehensively analysed using ORFfinder online server (https://www.ncbi.nlm.nih.gov/orffinder/, accessed on 1 January 2025). Conserved protein domains were predicted using a combination of databases, including the Protein Data Bank (PDB) (https://www.rcsb.org/, accessed on 1 January 2025) and the Pfam database (https://www.ebi.ac.uk/interpro/, accessed on 1 January 2025). Specifically, the pfam_search.pl script was employed to scan for known domain motifs within the viral sequences, and HHblits (version 3.3.0) was used for more sensitive and in-depth sequence alignments, improving the accuracy of domain prediction by Hidden Markov models (HMMs). The predicted ORFs and protein domains were then visualized using the R package gggenes (version 0.5.1).

### 2.5. Amino Acid Similarity Comparison

Reference sequences of classified iflaviruses and highly homologous sequences were retrieved from GenBank and the ICTV database [22]. Pairwise sequence alignment and identity calculation [23] were constructed based on the amino acid (aa) sequences of the predicted conserved capsid protein (CP) region, RNA-dependent RNA polymerase (RdRp), and complete polyprotein of the selected iflaviruses. Sequence alignment was conducted using MAFFT (version 7.525) [24], and sequence similarity was calculated based on the percentage of amino acid identity.

### 2.6. 3D Structure Prediction

The polyproteins of iflaviruses were submitted to AlphaFold (https://alphafoldserver.com/, accessed on 1 January 2025) for 3D structure prediction. The reliability of the model was assessed using pLDDT (predicted Local Distance Difference Test) scores, with all predictions performed using default settings. pLDDT scores > 70 indicate high confidence in the predicted structure, while scores < 50 suggest low reliability. To evaluate the structural similarity and variability among the predicted protein structures, pairwise alignments were conducted using the TM-align tool [25].

### 2.7. Phylogenetic Analysis

Phylogenetic trees were constructed based on the amino acid sequences of RdRp, CP, and complete polyprotein, along with corresponding regions of representative viruses from various families within the order *Picornavirales*. Multiple sequence alignment was performed using MAFFT and retained conserved regions of longer fragments using Gblocks with the parameters (-b2 = 0.55 -b4 = 5 -b5 = h) (version 0.91b). The best amino acid substitution model was selected using ModelTest, and phylogenetic trees were constructed using the Maximum Likelihood (ML) algorithm by RAxML-NG (version 0.9.0) with 1000 bootstrap replicates [26]. The genus *Triatovirus* was used as an outgroup to root the trees.

## 3. Results

### 3.1. Three Iflaviruses Identified in Aleyrodes proletellus

A total of 97,954 contigs longer than 200 bp were obtained from the de novo assembly of clean reads. The cabbage whitefly species was identified as *A. proletellus*, with a 99.83% COI sequence similarity to the previously reported *A. proletellus* sequence (Accession Number: MK168804.1). Through a homology search against the local virus database, 16 contigs (comprising a total of 17,890 reads) from various viral families were identified. Among these, three nearly complete genomes were successfully identified and designated as Aleyrodes proletellus iflavirus 1 (APIV1, 9665 nt, 76.2× coverage) (Figure 1A), Aleyrodes proletellus iflavirus 2 (APIV2, 8780 nt, 81.7× coverage) (Figure 1B), and Aleyrodes proletellus 3 (APIV3, 8608 nt, 25.5× coverage) (Figure 1C). Comparison with the NCBI NR database revealed the highest sequence similarity of 38.71% between APIV1 and Bat faecal-associated iflavirus 2 (ON872543.1), while APIV2 and APIV3 exhibited highest sequence similarities of 47.08% (Novo Mesto iflavirus 1, OL472192.1) and 41.72% (Myrmica rubra picorna-like virus 8, MW314655.1), respectively (Table 1). These results indicate that the three viruses are likely new species. The three viral contigs were further verified by RT-PCR, followed by Sanger sequencing. There is an overlapping area between every two adjacent fragments for the three viruses to eliminate false positives (such as misassembled contigs, recombination, and others) (Appendix A). The genome sequences of APIV1 (9665 nt), APIV2 (8780 nt) and APIV3 (8608 nt) were subsequently submitted to GenBank, with the accession numbers PQ888994, PQ888995 and PQ888996, respectively.

### 3.2. Genome Structure of Three Novel Viruses

APIV1, APIV2 and APIV3 are predicted to contain a single ORF encoding polyproteins of 2891, 2851, and 2764 amino acids, respectively. InterProScan analysis revealed that both APIV1 and APIV2 harbor five conserved domains: a picornavirus-like capsid domain (Rhv), a CRPV_capsid superfamily domain (CRPV), an RNA helicase domain (Hel), a 3-chymotrypsin-like protease domain (Pro), and an RNA-dependent RNA polymerase domain (RdRp). In APIV1, the SPs comprise an Rhv domain (2737–3163 nt) and a CRPV domain (3709–4348 nt), while the NSPs include a Hel domain (5497–5818 nt), a Pro domain (nucleotides 7414–7717 nt), and an RdRp domain (7978–9364 nt) (Figure 1A). In APIV2, the structural proteins consist of an Rhv domain (857–1388 nt) and a CRPV domain (1511–2276 nt), whereas the non-structural proteins are represented by a Hel domain (4235–4559 nt), a Pro domain (6386–6956 nt), and an RdRp domain (7211–8600 nt) (Figure 1B). APIV3 is characterized by a single CRPV domain (372–879 nt) as its structural component, with non-structural proteins including a Hel domain (3057–3381 nt), a Pro domain (5283–5766 nt), and an RdRp domain (6039–7398 nt) (Figure 1C). Notably, the NSPs are arranged in the order Hel–Pro–RdRp, from the N- to the C-terminus. The Hel domain, belonging to superfamily 3 and associated with NTP binding, is characterized by a conserved motif A (2185–2192 nt: GxxGxGKS) [27]; the chymotrypsin-like cysteine protease domain, which resembles the 3C protease of picornaviruses, contains a cysteine protease motif (3495–3498 nt: GxCG) [28] and a substrate-binding motif (3515–3520 nt: GxHxxG) [27]; and the RdRp domain, a member of RNA polymerase superfamily I, harbors conserved motifs including motif V (3932–3943 nt: PSGx_3_Tx_3_N(S/T)_2_) and motif VI (3984–3987 nt: YGDD) [29] (Appendix A).

### 3.3. The Evolutionary Distances of APIV1–3 and Other Iflaviruses

To determine whether the three iflaviruses are independent virus species or merely different isolates of the same virus, their CP and RdRp were compared with ICTV-confirmed Iflaviruses using the SDT. The results revealed that these three iflavirus species have consistent similarities in the CP (Figure 2A) and RdRp (Figure 2B), with similarities of 43% (ON872543, Bat faecal-associated iflavirus 2), 40% (NC_002066 Sacbrood virus), and 37% (NC_002066, Sacbrood virus) compared to ICTV-recognized iflaviruses species. Based on ICTV criteria for new Iflaviruse species (intraspecies differentiation requiring >90% capsid protein amino acid sequence similarity) (https://ictv.global/report/chapter/iflaviridae/iflaviridae/iflavirus, accessed on 1 January 2025), APIV1, APIV2, and APIV3 were confirmed as three distinct novel Iflaviruses species (Figure 2).

### 3.4. Phylogenetic Analysis of Three Iflaviruses

To confirm the authentic taxonomic status of the three iflaviruses, phylogenetic trees based on the CP (Figure 3A), RdRp (Figure 3B) and Polyprotein (Appendix A) sequences of representative members of the family *Iflaviridae* were constructed. Both the polyprotein and CP-based phylogenetic trees exhibited highly consistent evolutionary relationships, indicating similar conservation patterns in Iflaviruse evolution. APIV1, APIV2, and APIV3 clustered in three distinct evolutionary branches. APIV1 exhibited the closest phylogenetic relationship with Bat faecal-associated iflavirus 2, while APIV2 and APIV3 grouped with evolutionary branches that include Hubei picorna-like virus 40 (NC 032756.1). The high-confidence phylogenetic tree supports the presence of potentially undiscovered Iflaviruse diversity (Figure 3 and Appendix A).

### 3.5. Viral 3D Structure Comparison

Phylogenetic analysis revealed that the three viruses did not form a single cluster, suggesting that they may have distinct origins. To further explore the differences among them, we predicted the 3D structures of APIV1–3 and performed a comparative analysis. AlphaFold3 predictions of polyprotein structures yielded average pLDDT scores of 64.93, 70.69, and 61.64 for APIV1, APIV2, and APIV3, respectively. TM-align analysis of the 3D structures revealed highly conserved core regions but significant variations in the external loop regions. The comparison between APIV1 and APIV2 showed the most pronounced variations, with a root-mean-square deviation (RMSD) of 8.60 Å and a TM-score of 0.33784 (Figure 4A). The comparison between APIV1 and APIV3 exhibited slightly less variation, with an RMSD of 7.72 Å and a TM-score of 0.34955 (Figure 4B). The comparison between APIV2 and APIV3 demonstrated relatively higher structural similarity, with an RMSD of 7.64 Å and a TM-score of 0.36407 (Figure 4C).

## 4. Discussion

Hemipteran insects are major agricultural pests that can transmit a variety of plant viruses, leading to significant economic losses in crop production [30]. Recent studies have reported the identification and discovery, through high-throughput sequencing, of many new viruses (insect-specific viruses, ISVs) in vectors, such as leafhoppers, aphids, and whiteflies, that do not cause disease in hosts and usually have no economic impact on plants [14]. However, despite the close taxonomic status of cabbage whiteflies, the viruses they carry have not yet been effectively investigated. Here, three novel iflaviruses were identified in *A. proletella* using meta-transcriptomics technology for the first time: APIV1, APIV2, and APIV3.

Viruses in the family *Iflaviridae* are positive-sense, single-stranded RNA viruses that are widely distributed across arthropods [31]. According to the International Committee on Taxonomy of Viruses (ICTV), the virus consists of 16 species (https://ictv.global/report/chapter/iflaviridae/iflaviridae/iflavirus, accessed on 10 February 2025). Known hosts now include insects belonging to the orders Orthoptera [32], Hymenoptera [33,34], Lepidoptera [35,36], Hemiptera [37], Odonata [17], and Diptera [38]. *Iflavirus* is primarily transmitted horizontally (such as food contamination) and vertically (from mother to offspring); host infection may be asymptomatic or lead to severe pathology. Deformed wing virus (DWV)-affected bees emerged twice as slowly and had a 30% higher mortality rate compared to clinically normal bees [39]. Bombyx mori Infectious flacherie virus (BmIFV) causes flacherie-like symptoms and is known to result in crop losses of up to 40% [40]. However, Nilaparvata lugens honeydew virus 1 (NLHV1) does not induce symptoms in the host [16]. Here, we identified three novel members, APIV1–3, in *A. proletella*, a species whose virome has not been fully explored before, suggesting that the diversity of cabbage whitefly viruses may be significantly underestimated. We used InterProScan to search for conserved domains in the amino acid sequence and identified multiple viral coat domains at the N-terminal end, as well as a helicase and RdRP domain at the C-terminal end (Figure 1). The closest relative to APIV1 in the phylogenetic tree is Bat faecal-associated iflavirus 2 (ON872543.1), which is found in bat faeces and may be associated with the bat’s diet [41]. APIV2 was clustered with Novo Mesto iflavirus 1 (OL472192.1), the virus reported in the tomato and weed virome project [42], and APIV3 related to Myrmica rubra picorna-like virus 8 (MW314655.1), a virus found in ants [43]. Currently, the transmission pathway, infection mechanisms, and effects on the host species of APIV1–3 remain unclear and require further investigation.

This study employed AlphaFold3 to gain deeper insights into the structural characteristics of these viruses. Structural modelling revealed conserved core regions and highly variable external loops, with RMSD values ranging from 7.64 to 8.60 Å (Figure 4). These findings show some structural flexibility, which may play a crucial role in host adaptation and interaction [44]. The conserved core regions, characterized by high-confidence pLDDT scores and essential viral functions, such as replication and assembly, ensure the fundamental viability of the viruses [45]. In contrast, the variable regions, which exhibit greater structural flexibility, likely mediate host-specific interactions and immune evasion, enabling the viruses to spread and adapt to new hosts [46].

In summary, this study identified three new viruses in *A. proletella*. Based on phylogenetic analysis, they were classified into the family *Iflaviridae*. Our research contributes to a better understanding of the viral diversity in *A. proletella*, offering valuable insights into the coevolution between iflaviruses and their host. Further studies are essential to understanding the possible functions of these viruses and their impact on agricultural productivity.

## 5. Conclusions

This study is the first to use meta-transcriptomics to identify three novel viruses in the cabbage whitefly (*A. proletella*). By analysing the genomic structure and the evolutionary relationships of these viruses, we determined that the three viruses all belong to the family *Iflaviridae* and designated them as APIV1, APIV2, and APIV3. Further structural predictions of the polyprotein using AlphaFold3 revealed that while the overall structural framework of these viruses is conserved, significant variations exist, particularly in the external loop regions. Currently, the understanding of structural variability in *Iflavirus* remains limited. These findings provide valuable insights into the composition and function of the virome in the cabbage whitefly, as well as virus–host interactions.

## Figures and Tables

**Figure 1 insects-16-00335-f001:**
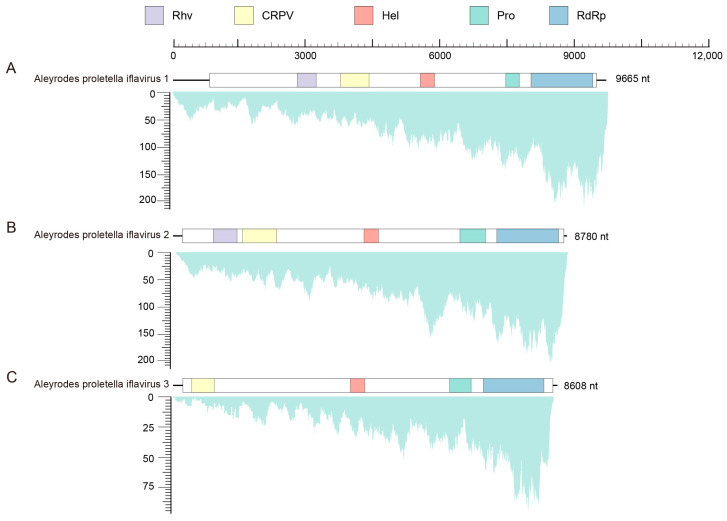
Genome structure and transcriptome raw read coverage of AIPV1 (**A**), AIPV2 (**B**) and AIPV3 (**C**). The white frame represents ORFs. Blocks of different colours represent different domains of the polyprotein. Purple: Rhv, picornavirus-like capsid domain; Yellow: CRPV, CRPV_capsid super family domain; Red: Hel, RNA helicase domain; Green: Pro, 3-chymotrypsin-like protease domain; Blue: RdRp, RNA-dependent RNA polymerase domain.

**Figure 2 insects-16-00335-f002:**
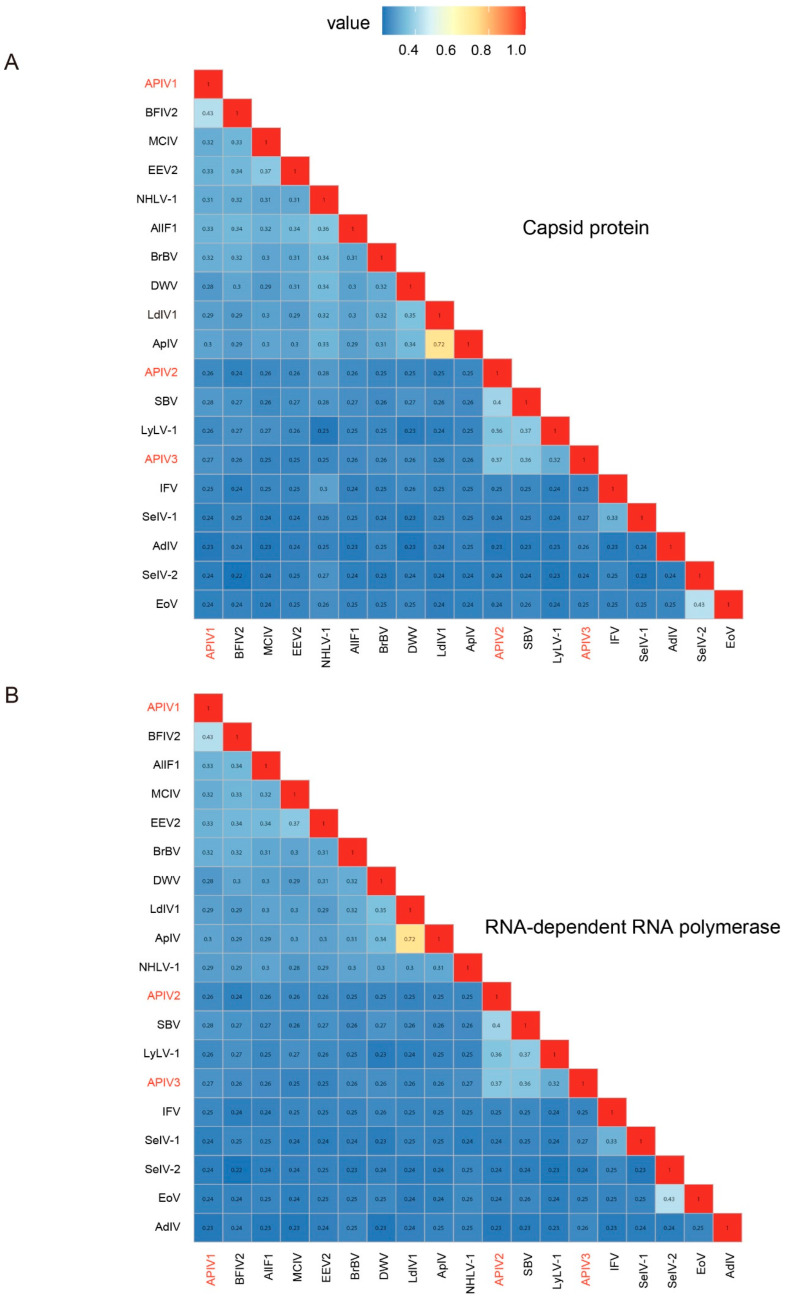
Pairwise distances between CP (**A**) and RdRp (**B**) sequences among the iflaviruses identified in cabbage whiteflies. Through the predicted CP and RdRp domains, a pairwise distance matrix was constructed using MAFFT for the three newly discovered iflaviruses and homologous iflavirus sequences included in ICTV and NCBI GenBank using ggplot2. The new virus sequences identified in this study were highlighted in red. EoV: Ectropis obliqua picorna-like virus, SeIV-2: Spodoptera exigua iflavirus 2, AdIV: Acheta domesticus iflavirus, SeIV-1: Spodoptera exigua iflavirus 1, IFV: Infectious flacherie virus, LyLV-1: Lygus lineolaris virus 1, SBV: Sacbrood virus, ApIV: Antheraea pernyi iflavirus, LdIV1: Lymantria dispar iflavirus 1, DWV: Deformed wing virus, BrBV: Brevicoryne brassicae virus—UK, NHLV-1: Nilaparvata lugens honeydew virus 1, DcPV: Dinocampus coccinellae paralysis virus, VDV-2: Varroa destructor virus 2, SBPV: Slow bee paralysis virus, EEV2: Exitianus exitiosus virus 2, MCIV: Medvezhye Chrysops Ifla−like virus, BFIV2: Bat faecal-associated iflavirus 2, AlIF1: Aulacophora lewisii iflavirus 1. The viral Accession numbers and name are listed in Appendix A.

**Figure 3 insects-16-00335-f003:**
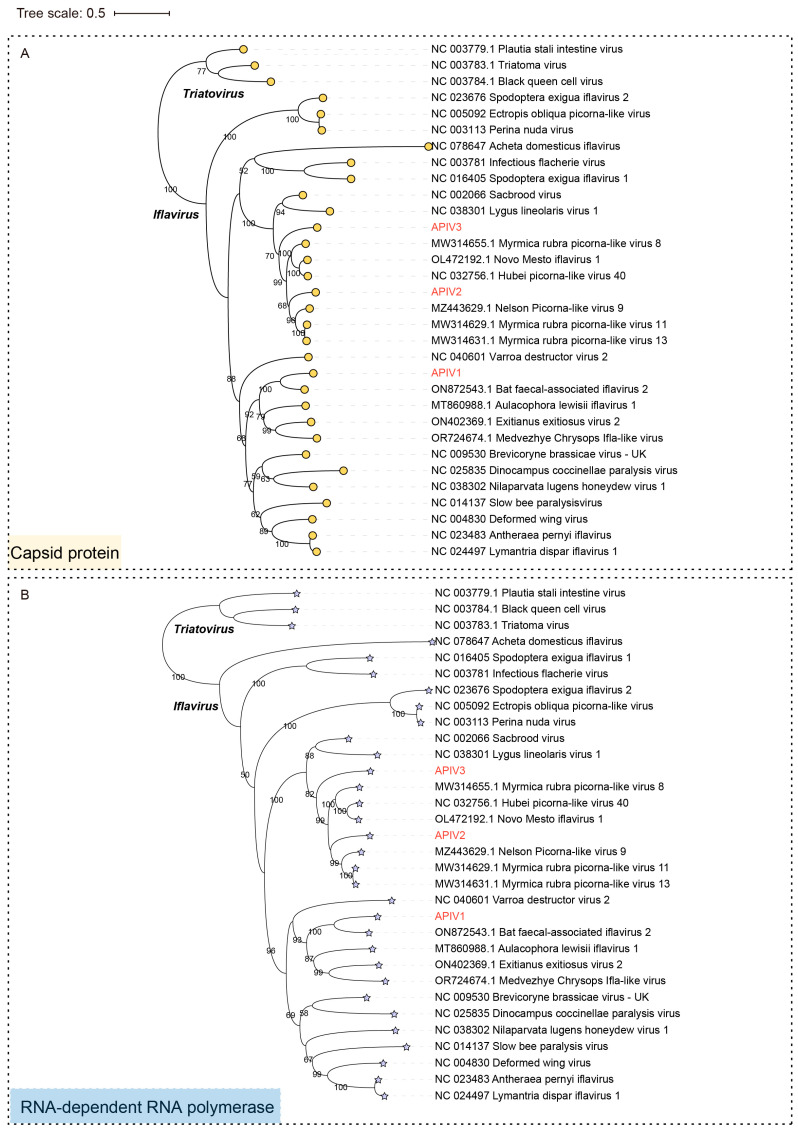
Phylogenetic trees based on the CP (indicated by circular markers) (**A**) and RdRp (indicated by asterisks) (**B**) of the three iflaviruses identified in cabbage whiteflies. APIV1, APIV2 and APIV3 are highlighted in red font. Nodes with bootstrap values > 50% are marked.

**Figure 4 insects-16-00335-f004:**
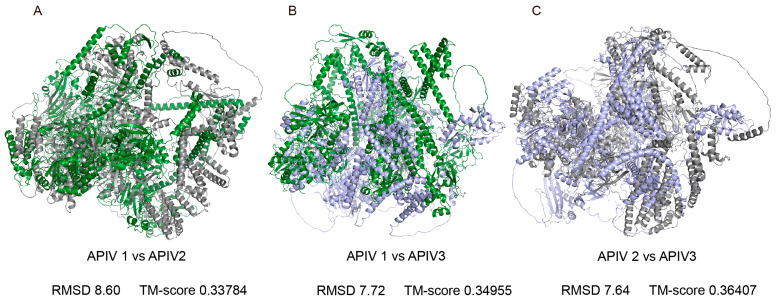
Structural comparison of APIV1, APIV2 and APIV3 proteins and highlighted conserved cores and divergent external loops. Superimposed structures of the three polyproteins: APIV1 (green), APIV2 (grey), and APIV3 (purple). ALPV1 is aligned with ALPV2 (**A**) ALPV1 is aligned with ALPV3 (**B**) ALPV2 is aligned with ALPV3 (**C**).

**Table 1 insects-16-00335-t001:** The detailed information about the three novel viruses identified in *Aleyrodes proletella*.

Virus Names	Accession	Length (nt)	Homologous Virus Accession	Closest Homologous Virus	Cover (%)	Identity (%)	Aligned Length	E-Value
Aleyrodes proletellus iflavirus 1 (APIV1)	PQ888994	9665	ON872543.1	Bat faecal-associated iflavirus 2	63%	38.71%	1873	0
Aleyrodes proletellus iflavirus 2 (APIV2)	PQ888995	8780	OL472192.1	Novo Mesto iflavirus 1	98%	47.08%	2849	0
Aleyrodes proletellus iflavirus 3 (APIV3)	PQ888996	8608	MW314655.1	Myrmica rubra picorna-like virus 8	54%	41.72%	1806	0

## Data Availability

The original contributions presented in this study are included in the article/Appendix A. Further inquiries can be directed to the corresponding authors.

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
