# Peer review of "Identification and Characterization of Three Novel Iflaviruses in the Cabbage Whitefly Aleyrodes proletella"

_insects, 2025, doi:10.3390/insects16040335_

Round 1
Reviewer 1 Report
Comments and Suggestions for Authors
In the reviewed manuscript, a group of scientists presents comparative transcriptomic analyses aimed at identifying viral diversity in Aleyrodes proletella, an economically important pest insect. The authors report the discovery of three new species of viruses belonging to the genus Iflavirus, characterize their genomes, and perform phylogenetic analyses to assess their placement within the Iflavirus genus. While the study addresses an interesting and relevant topic, the manuscript requires significant revision before it can be considered suitable for publication. The MS needs serious revision. The text includes overestimations and overgeneralizations (mainly concerning evolution and interactions with host, which actually were not studied). Phylogenetic analyses resulted in wrong similar gene trees, that should be revised. Most figures and captions need reconsideration. Discussion and Conclusions are written very poorly. The authors are requested to provide explanation for close similarity of the new Iflavirus with the one from Bat feces (ON872543). Additional remarks are given below.
19 each forming a distinct evolutionary cluster in the phylogenetic tree – please, revise this. A cluster should comprise at least 2 samples
21 and provide a foundation for future research and the development of potential biocontrol strategies – this is very broad/uncertain. Please, specify what exactly do you want to say.
27 (Hemiptera: Aleyrodidae) – no need to repeat this, it was already said above
29 providing new insights into viruses associated with the cabbage whitefly – redundant
34 evolutionary dynamics of A. proletella - ??? how exactly it was shown?
69 with iflaviruses identified exclusively in arthropods, primarily insects. – please, compare this with sequence ON872543 from your tree and address this correctly in this sentence
77 as well as the interactions between viruses and their host – did you really study the interactions?
76 vs 78 Moreover, this study addresses a gap in exploring viral diversity within the family Aleyrodidae – please, combine 76 and 78
82 with the kind help of De-Yin Ma (Xinjiang Agricultural University) – please, transfer this to Aknowledgements, “kind help” is redundant
95 The flited contigs - ??
139 Gblocks – please, specify which adjustment of this program was used? Stringent? Non-stringent? Etc
142 The family Solemoviridae was used as an outgroup to root the trees – I do not see members of this family in the given tress. Please, revise this
Table 1 this table is largely discussed in the text, please, consider transferring it to Supplement
Fig2 please, explain what are the red squares (diagonal)?
215 CP (Figure 3A) and RdRp – why only these 2 genes were chosen? Please, provide all gene trees for all genes
Figure 3: please revise the trees: they are identical with identical branch supports which is definitely a mistake. Please, indicate out-groups unambiguously
230 suggesting distinct origins or divergent selective pressures – not clear. What do you mean saying this?
242-244 do you have any proofs to speculate this? Probably it is better to remove this from the text
Fig4 the colors are chosen suboptimal. Please, revise the color code and revise the caption.
251-257, 262-272 all these have been already said in the Introduction. Please, consider removing this from Discussion
The Discussion is written very poorly. It needs major revision.
The Conclusion is vague and uncertain. Please, provide a short stand-alone paragraph summarizing the most important results and give them in broader context
Author Response
Referee: 1 COMMENTS FOR AUTHORS
In the reviewed manuscript, a group of scientists presents comparative transcriptomic analyses aimed at identifying viral diversity in Aleyrodes proletella, an economically important pest insect. The authors report the discovery of three new species of viruses belonging to the genus Iflavirus, characterize their genomes, and perform phylogenetic analyses to assess their placement within the Iflavirus genus. While the study addresses an interesting and relevant topic, the manuscript requires significant revision before it can be considered suitable for publication. The MS needs serious revision. The text includes overestimations and overgeneralizations (mainly concerning evolution and interactions with host, which actually were not studied). Phylogenetic analyses resulted in wrong similar gene trees, that should be revised. Most figures and captions need reconsideration. Discussion and Conclusions are written very poorly. The authors are requested to provide explanation for close similarity of the new Iflavirus with the one from Bat feces (ON872543).
Response: We thanks for your comments and suggestions. We have carefully revised our manuscript addressing each of your comments/suggestions. The language has been polished and improved throughout of the manuscript to make it suitable for publication.
Additional remarks are given below:
- 19 each forming a distinct evolutionary cluster in the phylogenetic tree – please, revise this. A cluster should comprise at least 2 samples.
Response: We apologize for the confusion. The sentence has been modified for the clarity (Please refer to lines 19-20).
- 21 and provide a foundation for future research and the development of potential biocontrol strategies – this is very broad/uncertain. Please, specify what exactly do you want to say.
Response: Thank you for your comment. With the increasing interest in biological control methods, exploring viral resources in insect-particularly those carried by agricultural pests-can contribute to the development of biopesticides for pest management. This study focuses on identifying the viral composition of the cabbage whitefly, an agricultural pest, with the aim of providing valuable research data for related biological control strategies. We have refined the relevant sections of the revised manuscript accordingly (please refer to lines 21-22).
- 27 (Hemiptera: Aleyrodidae) – no need to repeat this, it was already said above.
Response: Done (please refer to line 28).
- 29 providing new insights into viruses associated with the cabbage whitefly – redundant
Response: Done (please refer to line 30).
- 34 evolutionary dynamics of A. proletella - ??? how exactly it was shown?
Response: Thank you for your valuable comments. We have revised it (Line 34).
- 69 with iflaviruses identified exclusively in arthropods, primarily insects. – please, compare this with sequence ON872543 from your tree and address this correctly in this sentence.
Response: Thank you for your comments. According to the ICTV Virus Taxonomy Profile (ICTV Virus Taxonomy Profile: Iflaviridae) (https://ictv.global/report/chapter/iflaviridae/iflaviridae), all members infect arthropod hosts with the majority infecting insects. According to the description in the article (Faecal virome of the Australian grey-headed flying fox from urban/suburban environments contains novel coronaviruses, retroviruses and sapoviruses), ON872543 was identified in bat feces and may be associated with the bat's diet (Please refer to lines 285-286).
- 77 as well as the interactions between viruses and their host – did you really study the interactions?
Response: We sincerely appreciate the valuable comments. We are not currently investigating the interaction between viruses and A. proletella and have revised the relevant content in the manuscript accordingly (please refer to lines 81-83).
- 76 vs 78 Moreover, this study addresses a gap in exploring viral diversity within the family Aleyrodidae – please, combine 76 and 78.
Response: Thank you for your comment. We have modified the description in manuscript (Lines 81-83).
- 82 with the kind help of De-Yin Ma (Xinjiang Agricultural University) – please, transfer this to Aknowledgements, “kind help” is redundant.\
Response: Done (Please see lines 86-88 and 320-321).
- 95 The flited contigs - ??
Response: Thank you again for your comments. To eliminate the impact of some contaminated sequences on virus identification results, only contigs longer than 200 bp were analyzed in this study (Lines 98-100).
- 139 Gblocks – please, specify which adjustment of this program was used? Stringent? Non-stringent? Etc
Response: We agree with the reviewer's comments. Multiple alignment sequence was retained conserved regions of longer fragments using Gblocks with the parameters (-b2=0.55 -b4=5 -b5=h). This has been corrected in the revised manuscript (Please see lines 143-144).
12.142 The family Solemoviridae was used as an outgroup to root the trees – I do not see members of this family in the given tress. Please, revise this
Response: We apologize for this and have revised in the lines 147-148 and Figure 3A.
13.Table 1 this table is largely discussed in the text, please, consider transferring it to Supplement
Response: Thank you for your comment. Table 1 provides detailed information about the three novel viruses identified in A. proletella. We consider it should be designated as Table 1.
14.Fig2 please, explain what are the red squares (diagonal)?
Response: Figure 2 illustrates the viral amino acid similarity matrix based on pairwise sequence alignment and identity calculation. A pairwise comparison is conducted between the two viruses corresponding to the horizontal and vertical directions. Red squares indicate self-comparisons, where the identity value is 1.
- 215 CP (Figure 3A) and RdRp – why only these 2 genes were chosen? Please, provide all gene trees for all genes
Response: Thank you for your comment. RdRp, commonly used as a marker gene for classifying RNA viruses, is located in the non-structural protein region, while CP, recommended by the ICTV for determining iflavirus taxonomy, is found in the structural protein region. According to your suggestion, we have constructed the phylogenetic tree of the polyprotein in Figure S2. Notably, the classification results based on CP, RdRp, and polyprotein are nearly identical (Lines 227-229, 235-236 and revised Figure S2).
- Figure 3: please revise the trees: they are identical with identical branch supports which is definitely a mistake. Please, indicate out-groups unambiguously
Response: We are really sorry for our careless mistakes and sincerely appreciate your careful reading. Following your comments, we have corrected the Figure 3 (Please see revised Figure 3).
17.230 pressures – not clear. What do you mean saying this?
Response: Thank you for your comment. Although our study identified three iflaviruses within a population of cabbage whiteflies, these three iflaviruses did not cluster on a single branch and even showed significant evolutionary distances, suggesting that they may have distinct origins. We have revised the manuscript (Please see Line 243).
- 242-244 do you have any proofs to speculate this? Probably it is better to remove this from the text
Response: Thank you for your careful inspection, we have deleted the description of the relevant sections.
- Fig4 the colors are chosen suboptimal. Please, revise the color code and revise the caption.
Response: We appreciate your suggestions and have revised the Figure 4 and the manuscript (Please see revised Figure 4 and lines 255-257).
- 251-257, 262-272 all these have been already said in the Introduction. Please, consider removing this from Discussion
Response: Thank you for your comment. The sentence has been modified (Lines 259-264).
- The Discussion is written very poorly. It needs major revision.
Response: We have revised this part according to your suggestions. Please see the revised manuscript about Discussion section.
- The Conclusion is vague and uncertain. Please, provide a short stand-alone paragraph summarizing the most important results and give them in broader context
Response: We have revised this part according to your suggestions. Please see the revised manuscript about Conclusion section. Specific modifications in lines 307-315.
Reviewer 2 Report
Comments and Suggestions for Authors
In this manuscript, the authors report examining a population of the cabbage whitefly using meta-transcriptomic analysis to identify three novel iflaviruses. They present good evidence to describe the new viruses. The work could be useful for future work on biological control of the pest. I recommend that the work be published after some minor revision. Specific comments:
Title, lines 1-3. I think “cabbage whitefly” needs to be capitalised in the title
Abstract, lines 23-36. Some detail on how the whitefly samples were collected needs to go into the abstract and in the methods section for the paper. The sampling seems somewhat limited, perhaps with only one population being sampled. This low level of sampling has implications for the potential of the virus.
Introduction, lines 74-75. Its not immediately clear why AlphaFold was used here. Could more detail please be added on why it might be useful? Has it been beneficially used with viral protein structure modelling previously? Perhaps more information could be presented in the methods on the use of AlphaFold as well.
Methods, lines 82-85. As for the introduction, a bit more information on the source and sampling of the whitefly would be useful here. Otherewise the methods seem to be clearly presented with sufficient detail. Some of the text, such as in the website addresses, needs the font to be adjusted.
Table 1, line 165. The title or caption for the table appears twice. One of these titles should be removed. More detail is needed in the caption, explaining the different parts of the table. The coverage for APIV2 was 98% with the Novo Mesto Iflavirus 1, which seems very high and very related?
Results, line 203. You should include a citation for the ICTV criteria in the results.
Figure 2, line 206. There are many abbreviations for viral names on the figure. These abbreviations with full viral names should be shown somewhere, perhaps in the caption.
Discussion, lines 250-306. The discussion is short, but fine. Please italicise the insect genus and species names throughout. Is the ICTV citation on lines 263-264 appropriate for the journal? I’d also recommend noting in the discussion that the viruses were obtained from a small sample, perhaps from a ‘healthy’ population of whitefly? There are likely to be other viruses in the environment and perhaps more lethal viruses for the whitefly.
Discussion, lines 285-286. This section of the discussion on AlphaFold was useful. Is the level of structural variation, that the authors suggest displays ‘significant structural plasticity’, typical of Iflaviruses? Or don’t we know? Seems like a big jump to suggest plasticity?
References, lines 318-416. These seem approapriate.
Figure legends, lines 420-440. The figure legends and table heading should be removed. The supplementary material seems fine.
Author Response
Referee: 2 COMMENTS FOR AUTHORS
In this manuscript, the authors report examining a population of the cabbage whitefly using meta-transcriptomic analysis to identify three novel iflaviruses. They present good evidence to describe the new viruses. The work could be useful for future work on biological control of the pest. I recommend that the work be published after some minor revision.
Response: We appreciate your insightful comments and suggestions. The revised manuscript was carefully prepared to enhance the clarity for all of your points addressed.
Specific comments:
- Title, lines 1-3. I think “cabbage whitefly” needs to be capitalised in the title
Response: We sincerely apologize for the error in our previous manuscript and this has been corrected in the revised manuscript (Please see line 3).
- Abstract, lines 23-36. Some detail on how the whitefly samples were collected needs to go into the abstract and in the methods section for the paper. The sampling seems somewhat limited, perhaps with only one population being sampled. This low level of sampling has implications for the potential of the virus.
Response: Thank you for your comment. We have added the necessary sampling information into the Abstract and corresponding Materials and Methods (Please see lines 26-27, 86-88).
- Introduction, lines 74-75. Its not immediately clear why AlphaFold was used here. Could more detail please be added on why it might be useful? Has it been beneficially used with viral protein structure modelling previously? Perhaps more information could be presented in the methods on the use of AlphaFold as well.
Response: We appreciate your insightful comments and suggestions. We have added the necessary information to the Materials and Methods (Please see lines 60-64).
- Methods, lines 82-85. As for the introduction, a bit more information on the source and sampling of the whitefly would be useful here. Otherewise the methods seem to be clearly presented with sufficient detail. Some of the text, such as in the website addresses, needs the font to be adjusted.
Response: Thank you for your comment. We have added the necessary sampling information into the Materials and Methods (Please see lines 86-88). And the font of the website address has also been adjusted accordingly (Please see lines 209,269-271).
- Table 1, line 165. The title or caption for the table appears twice. One of these titles should be removed. More detail is needed in the caption, explaining the different parts of the table. The coverage for APIV2 was 98% with the Novo Mesto Iflavirus 1, which seems very high and very related?
Response: Thank you for your correction. We have modified Ttable1 (Please refer to the revised Table 1 and line 170). Additionally, although APIV2 shares a certain coverage with Novo Mesto iflavirus 1 (OL472192.1), their amino acid similarity is only 47.08%. Moreover, they are positioned in distinct branches of the phylogenetic tree, suggesting a considerable evolutionary divergence.
- Results, line 203. You should include a citation for the ICTV criteria in the results.
Response: We appreciate your insightful comments and suggestions. As suggested, we have added the necessary citation in the revised manuscript (Please refer to Lines 209-210).
- Figure 2, line 206. There are many abbreviations for viral names on the figure. These abbreviations with full viral names should be shown somewhere, perhaps in the caption.
Response: Thank you for your valuable comments. The viral accession numbers and name are listed in Supplementary Table S2. According to your suggestion, we have also added relevant figure legends in the revised manuscript (Please see lines 217-224).
- Discussion, lines 250-306. The discussion is short, but fine. Please italicise the insect genus and species names throughout. Is the ICTV citation on lines 263-264 appropriate for the journal? I’d also recommend noting in the discussion that the viruses were obtained from a small sample, perhaps from a ‘healthy’ population of whitefly? There are likely to be other viruses in the environment and perhaps more lethal viruses for the whitefly.
Response: Thank you for your valuable comments. We have corrected the part about italics (Please refer to line 272) and the font of the website address has also been adjusted accordingly (Lines 269-271). As you mentioned, we have identified three novel viruses solely from A. proletella collected at Xinjiang Agricultural University, suggesting that the diversity of whitefly viruses may be significantly underestimated. In our future research, we plan to expand the sampling range, identify additional viruses, and assess the potential risks of previously unreported pathogenic viruses. We have also incorporated additions into the revised manuscript (Please see lines 279-281).
- Discussion, lines 285-286. This section of the discussion on AlphaFold was useful. Is the level of structural variation, that the authors suggest displays ‘significant structural plasticity’, typical of Iflaviruses? Or don’t we know? Seems like a big jump to suggest plasticity?
Response: Thank you for your comment. We used AlphaFold to predict the structure and calculated the pLDDT scores (64.93, 70.69, and 61.64, respectively). These values are moderate, indicating that the relevant regions may have some flexibility or prediction uncertainty. These pLDDT values do not clearly support the conclusion of “significant structural plasticity”. Therefore, in the revised manuscript, we will adjust the wording, changing “significant structural plasticity” to “some structural flexibility” to more accurately describe the variability of the predicted structure (Lines 289-290). And currently, the understanding of structural variability in Iflavirus remains limited, out findings also provide a foundation for understanding the structural changes of iflavirus.
- References, lines 318-416. These seem approapriate.
Response: Thank you very much for your encouragement.
- Figure legends, lines 420-440. The figure legends and table heading should be removed. The supplementary material seems fine.
Response: Thank you for your valuable comments and we have removed figure legends and table heading, please see the revised manuscript about Figure Legends section.
Round 2
Reviewer 1 Report
Comments and Suggestions for Authors
The authors revised carefully the MS and acquired most remarks by the reviewer. The MS looks better, and needs only minor final polishing. The authors are requested to read the whole text again to check it for typos and linguistic inconsistensies.
267 vs 302 remove repetition
302 A. proletella - italic, check in the entire text
262 Traditional agricultural pests are more attention - what does it mean?
Author Response
Comments and Suggestions for Authors:
Reviewer 1
The authors revised carefully the MS and acquired most remarks by the reviewer. The MS looks better, and needs only minor final polishing. The authors are requested to read the whole text again to check it for typos and linguistic inconsistensies.
Response: We sincerely appreciate the insightful comments and suggestions provided by reviewer 1. We have carefully revised the manuscript to enhance its clarity and suitability for publication. The revised content has been highlighted in yellow.
Additional remarks are given below:
- 267 vs 302 remove repetition
Response: Thank you for your comment. We consider that line 267 effectively summarizes the results of this experiment and also serving as a transition to the subsequent discussion on the family Iflaviridae. Therefore, we have retained this sentence. Additionally, we have made detailed modifications to line 302.
- 302 A. proletella - italic, check in the entire text
Response: Thank you for your comment. We have carefully reviewed the manuscript and made the necessary revisions accordingly.
- 262 Traditional agricultural pests are more attention - what does it mean?
Response: Thank you for your comment. We have revised the sentence, please see lines 259-263.